# Novel Hybrid Nanoparticles: Synthesis, Functionalization, Characterization, and Their Application in the Uptake of Scandium (III)Ions from Aqueous Media

**DOI:** 10.3390/ma13245727

**Published:** 2020-12-15

**Authors:** Ali Dawood Salman, Tatjána Juzsakova, Moayed G. Jalhoom, Cuong Le Phuoc, Saja Mohsen, Thamer Adnan Abdullah, Balázs Zsirka, Igor Cretescu, Endre Domokos, Catalina Daniela Stan

**Affiliations:** 1Laboratory for Surface and Nanostructures (LASUNA), Faculty of Engineering, University of Pannonia, 8200 Veszprém, Hungary; ali.dawood@mk.uni-pannon.hu (A.D.S.); yuzhakova@almos.uni-pannon.hu (T.J.); thamer.abdullah@mk.uni-pannon.hu (T.A.A.); zsirkab@almos.uni-pannon.hu (B.Z.); domokose@uni-pannon.hu (E.D.); 2Department of Chemical and Petroleum Refining Engineering, Basra University, Basra 61004, Iraq; 3Department of Production Engineering and Metallurgy, University of Technology, Baghdad 10001, Iraq; moayyedgassid@yahoo.com; 4Department of Environmental Management, Faculty of Environment, The University of DaNang- University of Science and Technology, DaNang 550000, Vietnam; lpcuong@dut.udn.vn; 5Nanotechnology and Advanced Materials Research Center, University of Technology, Baghdad 10001, Iraq; 11659@uotechnology.edu.iq; 6Faculty Chemical Engineering and Environmental Protection, “Gheorghe Asachi” Technical University of Iasi, 73, Blvd. D. Mangeron, 700050 Iasi, Romania; 7Department of Drug Industry and Pharmaceutical Biotechnology, “Grigore T. Popa” University of Medicine and Pharmacy, 700028 Iasi, Romania; catalinastan68@yahoo.com

**Keywords:** silica nanoparticles, molecular recognition, K2.2.2, scandium, rice husk

## Abstract

The aim of this study was to prepare novel supramolecular hybrid nanoparticles (HNPs) that can selectively separate and recover scandium metal ions, Sc(III), from an aqueous phase based on molecular recognition technology (MRT). Moreover, this approach is fully compatible with green chemistry principles. In this work, natural amorphous silica (SiO_2_) nanoparticles were prepared by a precipitation method from Iraqi rice husk (RH) followed by surface modification with 3-amino-propyl triethoxysilane (APTES) as coupling agent and Kryptofix 2.2.2 (K2.2.2) as polycyclic ligand. To evaluate the potential of the hybrid nanoparticles, the prepared HNPs were used for the solid–liquid extraction of scandium, Sc(III), ions from model solutions due to the fact that K2.2.2 are polycyclic molecules. These polycyclic molecules are able to encapsulate cations according to the corresponding cavity size with the ionic radius of metal by providing a higher protection due their cage-like structures. Moreover, the authors set the objectives to design a high-technology process using these HNPs and to develop a Sc recovery method from the aqueous model solution prior to employing it in industrial applications, e.g., for Sc recovery from red mud leachate. The concentrations of Sc model solutions were investigated using the UV-Vis spectrophotometer technique. Different characterization techniques were used including scanning electron microscope (SEM), atomic force microscopy (AFM), Brunauer–Emmett–Teller (BET), X-ray diffraction (XRD), X-ray fluorescence (XRF), and Fourier transform infrared (FTIR). The extraction efficiency of Sc varied from 81.3% to 96.7%. Moreover, the complexed Sc ions were efficiently recovered by HCl with 0.1 mol/L concentration. The stripping ratios of Sc obtained ranged from 93.1% to 97.8%.

## 1. Introduction

Attention has been paid lately to designing and preparing new macrocyclic host compounds because of their important role in nanoscience and supramolecular chemistry. For decades, nanomaterials have attracted great interest in various fields of science because of their smart chemical and physical properties, and these characteristics make them differ greatly from ordinary materials [1,2,3,4]. The amorphous silica nanoparticles prepared from rice husk (RH) have very important properties that qualify them to be excellent support materials for different ligands. This can be due to their high purity, high surface area, the great grafting of the dense functional groups onto the surface due to presence of dense hydroxyl group populations [5,6,7]. The combination of supramolecular chemistry and nanoparticles (NPs) can result in the creation of innovative hybrid supramolecular nanomaterials, which merge and improve the features of the two components, such as the catalytic, thermal, and electronic properties of NPs and the molecular identification of the macrocyclic hosts [8,9,10,11]. In 1978, supramolecular chemistry was defined by J.-M. Lehn as a new mainstream in chemistry, that is, a science dealing with investigations of the gathering of molecular building units associating by noncovalent forces [12,13,14]. A simple supramolecular substance is ordinarily created from the verification process between the two molecular construction pieces named receptor and substrate [7,15]. Supramolecular chemistry is a modern field of science, which tries to create simulated compounds that can be used for extremely special coupling required for particular molecular interactions [16,17,18]. Supramolecular compounds, such as crown ethers, cryptands, cavitands, cyclodextrins, cucurbit[n]urils (CB[n]), cyclophanes, rotaxanes, pillararenes and calixarenes, stay basic to supramolecular chemistry [19,20,21,22]. A complex constructed between a guest species and a cryptand is identified as a cryptate. The common generally applied cryptand is 4,7,13,16,21,24-hexaoxa-1,10-diazabicyclo [8.8.8] hexacosane, also termed as [2.2.2] cryptand or Kryptofix 2.2.2 [23,24]. The cryptands were improved by Jean-Marie Lehn, regarded as a type of crown ether derivative that has invited great interest due to its three-dimensional (3D) unique constructions and extra binding positions as presented in Figure 1.

The complexes they form with metal ions are thermodynamically much more stable than those formed by crown ethers [25,26,27]. Molecular recognition technology (MRT) has been attracting much attention lately. In the late 1960s, this technology began with the construction of crown ethers. That study evidenced assuring results on the particular complexation of cations and was crowned with the honor of the Nobel Prize in Chemistry in 1987 to Pedersen, Cram, and Lehn, three of the chief contributory scientists [10,28,29,30,31]. The special interaction between two or more molecules by noncovalent bonding, for example, hydrogen bonding, hydrophobic forces, metal coordination, halogen bonding, π–π interactions, van der Waals forces, and electrostatic and/or electromagnetic effects can be represented by the molecular recognition term [32,33,34]. The researchers proved that several supramolecular preparations can be prepared that exhibit molecular recognition features, where the guest species and the host species that are implicated in molecular recognition display molecular complementarity [35,36,37]. One of the most famous examples of these systems is crown ethers, which have the ability to integrate or selectively bind specific cations. Nonetheless, many simulated systems have since been designed [38]. “Host–guest” chemistry is the essential idea of MRT, where the creation of macrocyclic molecules with specific shapes and cavity sizes are appropriate for the specified cation diameter and, consequently, complexing characteristics [39,40,41]. The liquid-phase extraction process by macrocyclic compounds, termed LPE-MRT, has one disadvantage, namely the organic solvents that are used in separation processes [42]. This reason pushed the researchers to develop the solid-phase extraction (SPE) based on molecular recognition, termed SPE-MRT through the immobilization of macrocyclic compound ligands on the solid support materials [43,44]. The main advantages are that perpetual bonds can be taken from the binding of these ligands to solid supports such as polymeric materials or silica. Therefore, the system can be applied without loss of macrocyclic compounds and enables the collection of targeted ions by removing them with small volumes of solution [45,46,47]. Furthermore, the process can be implemented in a continuous system in fixed-bed columns [48]. In contrast with other conventional separation techniques, such as ion exchange, precipitation, and solvent extractions, the SPE-MRT process can overcome difficulties such as low purity of the outputs, low selectivity in the recovery, organic pollution of effluents, kinetic limitations, high space demands, and processing of low concentrated effluents. In addition to the above, the process is in accordance with the principles of green chemistry [49,50]. The MRT’s great selectivity compared to other techniques can be demonstrated by the multiple parameters that influence selectivity, including ionic radius, cation geometry requirements, ligand cavity dimensions, ring number, ligand donor/atom type and placement, and ring substituents. Consequently, ligands were joined into valuable forms such as membranes, beads, and columns [23,34]. This study ultimately led to the improvement of supramolecular hybrid nanomaterials based on host–connector–support materials in which a metal-selective ligand is joined to a solid support particle by a chemical linker, for example, a nanosilica (NS) surface. Moreover, the SPE-MRT process is compatible with clean chemistry principles to achieve efficient metal separations and recovery prior to employing it in industrial and commercial applications.

## 2. Materials and Methods

### 2.1. Materials

All chemicals were of analytical grade and used as received without further purification. 4,7,13,16,21,24-Hexaoxa-1,10-diazabicyclo [8.8.8] hexacosane (Kryptofix 2.2.2, 98%), 3-Aminopropyl-triethoxysilane (APTES, 98%), Ammonium hydroxide (NH4OH, 25%) and chloroacetic acid (ClCH_2_CO_2_H, 99.0%) were purchased from Sigma-Aldrich Co. (Munich, Germany). Ethanol (C_2_H_5_OH, 96%), acetic acid (CH₃COOH, 20% *v*/*v*), sodium hydroxide (NaOH, 97%), hydrochloric acid (HCl 37%), and scandium standard solution, 1000 mg/L Sc in diluted nitric acid (2%), AVS TITRINORM was used as standard for all measurements. All these were purchased from VWR Chemicals BDH Co. Xylenol orange (XO, 96%) was purchased from BDH Chemicals Ltd., Poole, UK. All aqueous solutions were prepared with distilled water.

### 2.2. Instrumentation

The silica nanoparticles were analyzed using the following instruments. A scanning electron microscope (TESCAN, Hillsboro, USA) was used for the surface studies. Fourier-transform infrared (FTIR) spectroscopic study spectra were carried out on a Bruker Vertex 70 spectrometer between 4000 and 400 cm^−1^. The XRD data were acquired on an X-ray diffractometer (Shimadzu S-6000, Kyoto, Japan). The Brunauer–Emmett–Teller (BET) method was applied using a Q Surf 1600 instrument, Mexico, USA. An atomic force microscope (AFM, Angstrom type, Scanning Probe Microscope, Advanced Inc., San Diego, CA, USA) was used as well. The determination of scandium concentrations in aqueous solutions was carried out using UV–Visible spectrophotometer (Shimadzu UV-1800, Kyoto, Japan) with a quartz cell of 1 cm.

### 2.3. Preparation of SiO_2_ Nanoparticles

To remove heavy impurities such as sand and/or dust, the rice husk (RH) was cleaned with distilled water, then the RH was dried for 5 h at 110 °C. The dried RH was refluxed for 90 min with 3 mol/L of hydrochloric acid (HCl). The treated RH was directly washed with warm distilled water. Subsequently, the obtained RH was dehydrated for 5 h at 110 °C. The treated RH was burned for 2 h at 700 °C under a heating rate of 5 °C/min. Subsequently, a white powder of rice husk ash (RHA) was obtained. Twenty grams of RHA were stirred in 160 mL NaOH (with a concentration of 2.5 mol/L). The aqueous solution was then refluxed for 3 h under continuous stirring. Equation (1) presents this reaction:(1)$$\begin{array}{c} 2\mathrm{NaOH}\\ \mathrm{Sodium}\;\mathrm{hydroxide} \end{array}+\begin{array}{c} {\mathrm{SiO}}_{2}\\ \mathrm{Ash} \end{array}\to \begin{array}{c} {\mathrm{Na}}_{2}{\mathrm{SiO}}_{3}\\ \mathrm{Sodium}\;\mathrm{silicate} \end{array}+\begin{array}{c} H_{2}O\\ \mathrm{Water} \end{array}$$

The produced solution was filtered with 40 mL of hot distilled water and the residue was washed. Afterward, the filtrate was cooled to 25 °C, 10 mol/L of sulfuric acid (H_2_SO_4_) was then added with constant stirring until the pH of the solution became about ~2 according to the following reaction, Equation (2):(2)$$\begin{array}{c} {\mathrm{Na}}_{2}{\mathrm{SiO}}_{3}\\ \mathrm{Sodium}\;\mathrm{silicate} \end{array}+\begin{array}{c} H_{2}{\mathrm{SO}}_{4}\\ \mathrm{Sulfuric}\;\mathrm{acid} \end{array}\to \begin{array}{c} {\mathrm{SiO}}_{2}\\ \mathrm{Ash} \end{array}+\begin{array}{c} {\mathrm{Na}}_{2}{\mathrm{SiO}}_{3}\\ \mathrm{Sodium}\;\mathrm{silicate} \end{array}+\begin{array}{c} H_{2}O\\ \mathrm{Water} \end{array}$$

Subsequently, a 25% ammonium hydroxide solution was continuously added until the pH was set to 8.5. Afterward, it was allowed to cool down to 250 °C for 3 h. Then, the product was refluxed for 4 h with 6 mol/L HCl and then washed frequently with distilled water to make it completely acid free. The obtained product was then dispersed under careful stirring in 2.5 mol/L NaOH, and the pH of the solution was set to 8. Following this, 10 mol/L of H_2_SO_4_ solution were also added. The obtained precipitate was, in fact, silica powder in nanoscale, which was washed with hot distilled water repeatedly to make it totally alkali free and then dried for 48 h at 50 °C [39]. The practical steps are described in the following block diagram in Figure 2.

### 2.4. Immobilized Ligands

The immobilization of Kryptofix 2.2.2 was carried out by the indirect coating protocol of SiO_2_ nanoparticles. The SiO_2_ nanoparticles were first functionalized with amino groups using APTES, since alkoxide groups of APTES were expected to react with OH groups on the silica surface. The immobilization method began by fixing the pH at 4.5 with a few drops of 1 mol/L acetic acid and followed by dispersing 100 mg of SiO_2_ nanoparticles in 40 mL EtOH/H_2_O solution at a ratio of 85:15 (% *v*/*v*). The beaker was put for 10 min in an ultrasonic bath. Afterward, 300 µL of APTES were added concurrently. The beaker’s content was placed under a refluxing system with stirring for 2 h at 75 °C. The modified particles obtained were separated from the mixture of reaction by centrifuge, washed once with ethanol and once with acetone also to remove the unreacted molecules, and then dried at 70 °C for 2 h to obtain the modified SiO_2_ nanoparticles. The modified nanoparticles SiO_2_-NH_2_ were functionalized with K2.2.2 through the formation of chemical bonds while the nanoparticles were dispersed by the sonication process and three drops of NH_4_OH in 0.01 mol/L of K2.2.2 solution with absolute ethanol solvent. The solution was then refluxed at 75 °C for 2 h under vigorous stirring. Finally, the nanoparticles were dried at 70 °C for 60 min, resulting in supramolecular species (host–connector–support) consisting of a solid support silica nanoparticle to which K2.2.2 (metal-selective ligand) was attached via a chemical linker arm. The linker arm was chemically joined to both the silica and the ligand, as shown in Figure 3.

### 2.5. Extraction and Back Extraction Experiments

The preliminary experiments were carried out in batch mode with different concentrations of scandium model solutions to evaluate the potential of the K2.2.2-functionalized silica nanoparticles. The results of the preliminary experiment showed that the extraction equilibrium could be reached within 15 min. The stock solutions of Sc(III) were prepared in different concentrations as described in Table 1. After the phase separation, the aqueous phase was separated by centrifuge.

The Sc concentration of the aqueous phases was measured using a spectrophotometric UV–VIS. instrument. To reach a solid–liquid ratio (s/l) of 1 g/L, the samples of 0.02 g of prepared hybrid nanomaterials were rigorously mixed with 20 mL liquid phase. To ensure that the concentration of the metals near the surface of the hybrid nanomaterial was roughly equivalent to that of the bulk phase and to reach maximum retention, the containers were agitated at 25 °C in an incubator with an agitation rate of 150 rpm. After 15 min, the solution was separated from the hybrid nanoparticles (HNPs) by centrifuging. Initial concentrations of aqueous phase and final concentrations were checked using a spectrophotometric UV–VIS instrument. The metal ion concentration in the solid phase can be obtained by using the equations for the mass balance. From Equation (3), the value of quantity of metal extracted per mass of hybrid nanomaterials was calculated, which is expressed as (q_e_), where C_0_ is the initial concentration and Ce is the equilibrium metal concentration in mg/L, m is the mass in g of hybrid nanomaterial, the volume (V) of the solution is expressed in L. The q_e_ is the extraction capacity at the equilibrium in mg/g. The extraction percentage, E%, was calculated according to Equation (4). The solid-phase separation contained the host–guest Sc ion complexes. In the following steps, we aimed for the recovery of Sc (III). Stripping or back extraction experiments were conducted by contacting the solid phase separated with 20 mL of 0.1 mol/L of hydrochloric acid solutions (stripping reagents) for 15 min in a shaking machine. The stripping percentage, S%, was calculated according to Equation (5), where C_st_ is the amount of Sc^3+^ released into aqueous solution (mg/L) and C_ex_ is the amount of Sc^3+^ extracted on the NPs (mg/L). All scandium concentrations were measured in duplicate by the UV method.
(3)$$q_{e}=\;\frac{\left(C_{0}-C_{e}\right)V}{m}$$
(4)$$E\%=\frac{\left(C_{0}-C_{e}\right)}{C_{0}}\times 100$$
(5)$$S\%=\;\frac{C_{\mathrm{st}.}}{C_{\mathrm{ex}.}}\times 100$$

### 2.6. Analytical Procedures

The determination of scandium concentrations in aqueous solutions was carried out by the spectrophotometric UV–VIS technique. The absorbance measurements were made using a UV-1800 spectrophotometer with a quartz cell. Xylenol orange (XO) reacts with scandium in lightly acidic medium to form a red-violet complex. This is the basis of a sensitive spectrophotometric identification method for the determination of scandium via the formation of bonding between metal ions and oxygen in xylenol orange. At 560 nm wavelength, xylenol orange absorbs imperceptibly, which is the maximum absorption of the complex. First, a 0.05 m % xylenol orange solution was prepared by dissolving XO in the distilled water. A standard scandium solution containing 1.00 mg Sc/mL was prepared. This solution was further diluted with water as required to produce different concentrations. Appropriate portions of NaOH (3 g) and ClCH_2_CO_2_H (15 mL) were taken to prepare a chloroacetate buffer of pH ~3, and the solution was diluted with water to 250 mL.

Then, 2.5 mL of the xylenol orange solution and 1 mL of the prepared buffer solution were added to a 25 mL standard flask containing a small volume of sample solution pH ~1 containing not more than 30 µg of Sc. The produced solution was diluted with 20 mL of water and the pH was adjusted to 2. This solution was transferred into a 25 mL standard flask and was made up to the mark with water and was left to stand for 10 min for full color development due to the addition of the reagent to the Sc. At 560 nm wavelength, the absorbance of the solution was measured using the UV–VIS method against a reagent blank solution [33].

## 3. Results and Discussion

### 3.1. Characterization

The chemical compositions of RHA and SiO_2_ nanoparticles were analyzed by X-ray fluorescence (XRF) as shown in Table 2. Since the metallic components have a strong impact on the particle size of the prepared SiO_2_ nanoparticles, the elimination of metallic impurities is therefore vitally essential. Moreover, there is a strong interaction between silica and the metallic ions, which results in a significant reduction in the surface area. Acid treatment was used to reduce the metallic impurities. It is favored to treat RHA with a tailored acidic solution to efficiently reduce these metallic contaminants, since the acid reacts chemically with the metal components and the reacted metals are leached from the acid solution during filtration. Thus, 3 mol/L of hydrochloric acid was used to obtain the optimum pretreatment process and to remove the metallic impurities. The SiO_2_ nanoparticles’ phase structure was investigated by X-ray diffraction. The XRD records confirmed the amorphous structure of the SiO_2_ nanoparticles, as shown in Figure 4, through the strong broad peaks of SiO_2_ nanoparticles centered at ≈ 22° (2θ), which is the characteristic amorphous structure of SiO_2_. Moreover, the absence of sharp peaks confirms the absence of the ordered crystalline structure.

The SEM images shown in Figure 5 illustrate the morphology, size, and distribution of the SiO_2_ nanoparticles. From the results, it can be observed that the SiO_2_ nanoparticles have an irregular shape and the spherical particles are agglomerated. This can be due to the forces acting among these particles (i.e., van der Waals forces). According to the Brunauer–Emmett–Teller (BET) method, the surface area of the prepared SiO_2_ nanoparticles was 293 m^2^/g. Consequently, the high surface area and dense hydroxyl groups of SiO_2_ play an important role in making it an excellent support. The morphology and the diameter distribution of the prepared SiO_2_ nanoparticles were investigated with three-dimensional surface profile using an atomic force microscope (AFM). The AFM records show the diameters of the prepared SiO_2_ nanoparticles, which are between 60 and 120 nm with 85.37 nm as an average diameter, as shown in Figure 6. The SiO_2_ nanoparticles are spherical and are agglomerated, as shown in Figure 7. The obtained results prove that the largest volume percentage of 14.06% concerning the particle size distribution was at 80.03 nm and the lowest volume percentage, 2.3%, was at 60 nm.

The FTIR spectra of the modified hybrid SiO_2_ nanoparticles are compared in Figure 8 to confirm the immobilization of organic groups on the surface of the SiO_2_ nanoparticles. Figure 8A shows the FTIR spectrum of the SiO_2_ nanoparticles. The peak around 1075 cm^−1^ indicates the Si–O–Si stretching vibration, while the shoulder at 1190 cm^−1^ is assigned to the Si–O vibration. Moreover, the deformation vibrations of these peaks (Si–O–Si and Si–O) can be seen at 456 cm^−1^ and 800 cm^−1^, respectively. A small amount of adsorbed water is present as is evident from the broad stretching (3000–3600 cm^−1^) and deformation (1633 cm^−1^) bands of water HOH.

After APTES modification (Figure 8B), changes are observed in the spectra. Presumably, due to its weak dipole-moment after surface adsorption, only very small intensities of N-H anti-symmetric and symmetric stretching are observed at 3358 and 3280 cm^-1^, respectively. The peak at 1564 cm^−1^ is assigned to the scissoring N–H_2_ deformation. The symmetric deformation of the NH_3_^+^ is observed at 1330 cm^−1^, while the antisymmetric counterpart is not clearly visible, as it overlaps with the water HOH deformation band at 1631 cm^−1^. The band at 1330 cm^−1^ could be attributed to the C–N stretching of the amine. Another clear evidence of successful APTES modification can be seen in the CH stretching region, where the antisymmetric and symmetric CH_2_ stretching are found at 2932 and 2863 cm^−1^, respectively. The CH_2_ deformation is located at 1440 cm^−1^. No major changes are observed for the SiO_2_ structure after modification.

The successful immobilization of K2.2.2 on the SiO_2_ nanoparticles can be confirmed by the FTIR spectrum in Figure 9C. The intense N–H bands are well observable at 3521, 3381, 1671, and 1615 cm^−1^, which are attributed to the N–H stretching and deformation. The C-N stretching vibration of K2.2.2 is found at 1475 cm^−1^. The C–O–C vibrations of ether are usually found in the 1310–1020 cm^−1^ region. A higher frequency shift in the C-H region of pure K2.2.2 (Figure 9D) from 3000–2820 cm^−1^ to 3050–2930 cm^−1^ indicates a modified chemical environment for the CH groups. The surface is covered with K2.2.2, and vibrations of the silica core are barely visible. However, the broader band at 1107 cm^−1^ might be assigned to the Si–O–Si stretching.

### 3.2. Mechanism of the Surface Modification

There are two principal approaches for indirectly joining macrocycles to the nanoparticles’ materials. The first one includes functionalization of the NPs’ surface by organic linking molecules, such as citrates [35], amines [34], or silanes [36,37], followed by the covalent linking of the macrocycles with the linkers. The second one includes covering the NPs with shell material, such as silica [41] and an organic polymer [38,39,40], and next embedding the macrocycles, non-covalently or covalently, inside the shell. The modification of the NPs’ surface with organic linkers is common, since the organic linkers give a broad spectrum of functionality for both types of covalent linkage. Ionic bonds can also be considered, as hydroxyls form amides or esters and amines, whereas thiols can form disulfide bridges and aldehydes can create imine linkages. Functionalization with ligands to solid surface support provides multiple reuses, reduces the demand for using high amounts of solvents, and opens the way for the advancement of eluents that considerably increases the efficacy of the separation method.

### 3.3. Extraction and Stripping/Back Extraction

The selectivity of macrocyclic compounds in extraction of the metal cation depends on different factors, including size and charge of the metal ion, nature of a donor atom, ligand cavity dimensions, ring number, ring substituents, diluents, and counter ion [42]. The results of preliminary SLE-MRT experiments show that the K2.2.2-modified SiO_2_ nanoparticles provided efficient extraction, as shown in Table 1. These results can be explained in that, on the one hand, the extraction ability of K2.2.2 for Sc(III) can be due to the molecular recognition and selective extraction of the Sc(III) cation and the excellent compatibility between the K2.2.2 macrocyclic compound ring sizes and the ionic radii of Sc ions. Correspondence can be seen between the extracted ion (Sc) radius and the cavity size of K2.2.2. On the other hand, the molecules of cryptands are three-dimensional structures and are able to encapsulate the guest ion by providing protection because of their cage-like structures or multidentate ligands for binding the guest ions. The stripping/back extraction of Sc(III) from K2.2.2-modified SiO_2_ nanoparticles was studied by hydrochloric acid (0.1 mol/L). According to the spectrophotometric determination of Sc, the results showed that the hydrochloric acid was very efficient in the process as most of the scandium was recovered, as shown in Table 1. Different studies have been conducted on solid-phase extraction (SPE) for the recovery of scandium ions such as ion-exchange adsorption and chelating resins, as shown in Table 3.

## 4. Conclusions

In this work, natural amorphous SiO_2_ nanoparticles were synthesized by a precipitation method from Iraqi rice husk and modified with 3-aminopropyl triethoxysilane (APTES) as chemical linker and 4,7,13,16,21,24-hexaoxa-1,10-diazabicyclo [8.8.8] hexacosane (Kryptofix 2.2.2) as supramolecular ligand. According to the SEM and AFM analyses, the SiO_2_ nanoparticles are spherical particles and have irregular shapes, and they agglomerate together in cluster form and the external diameters of the particles range between 60 and 120 nm with an average particle size of 85.37 nm. Moreover, the XRD, XRF, and BET studies confirmed the amorphous structure with high purity of 98.7% and surface area of 293 m^2^/g. According to the spectrophotometric determination of Sc, the extraction efficiency obtained ranged from 81.3% to 96.7%. Moreover, the stripping/back extraction efficiency with 0.1 mol/L of HCl ranged from 93.1% to 97.8%. It can be concluded that the loaded metal ion, scandium, can be efficiently recovered by HCl. The purpose of this study was to determine whether this kind of hybrid nanoparticles could combine the benefits of unique properties of nanomaterials and supramolecular macrocycle compounds to use in the separation and recovery of rare earth metals based on MRT. The combination of supramolecular macrocycles and NPs paves the way for the development of the production of hybrid nanomaterials.

## Figures and Tables

**Figure 1 materials-13-05727-f001:**
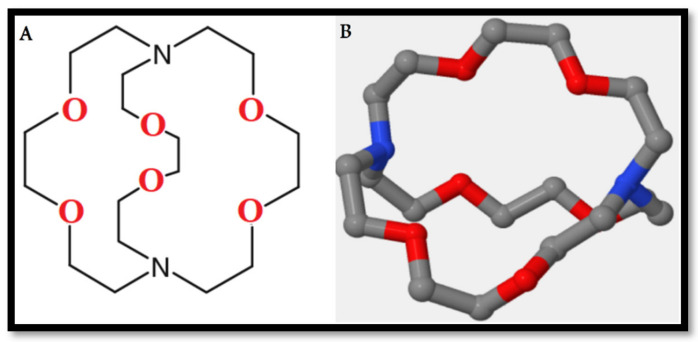
(**A**) Two-dimensional (2D) structure of K2.2.2, and (**B**) three-dimensional (3D) structure of K2.2.2.

**Figure 2 materials-13-05727-f002:**
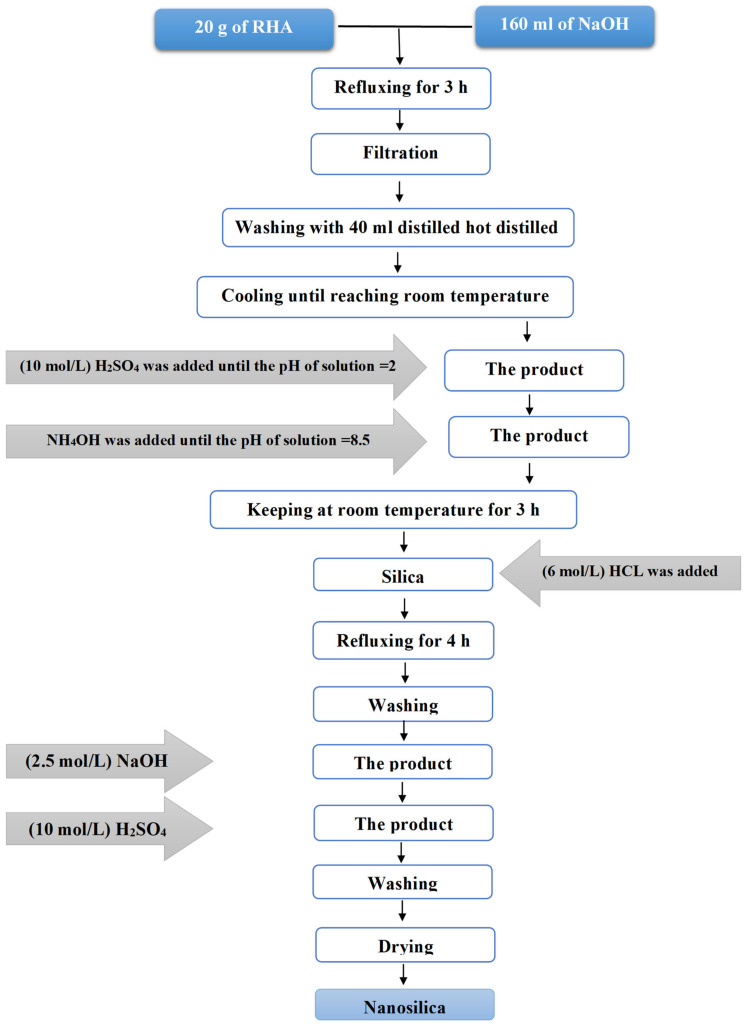
Block diagram of the preparation of nanosilica (NS) from rice husk (RH).

**Figure 3 materials-13-05727-f003:**
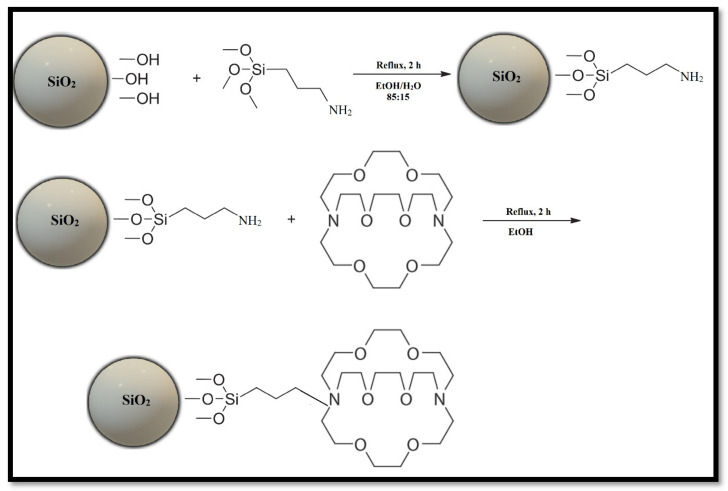
Representation of supramolecular hybrid nanomaterials based on host–connector–support materials.

**Figure 4 materials-13-05727-f004:**
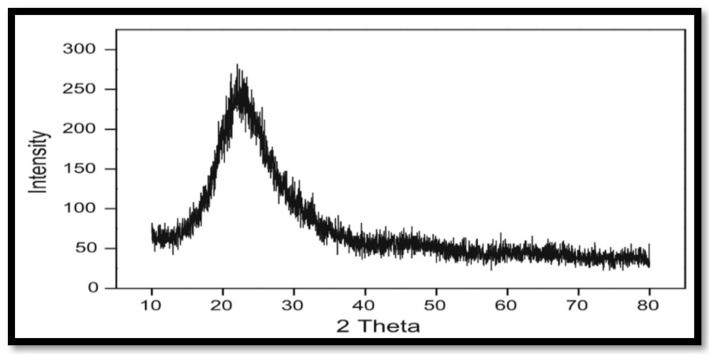
X-ray diffraction pattern of the SiO_2_ nanoparticles.

**Figure 5 materials-13-05727-f005:**
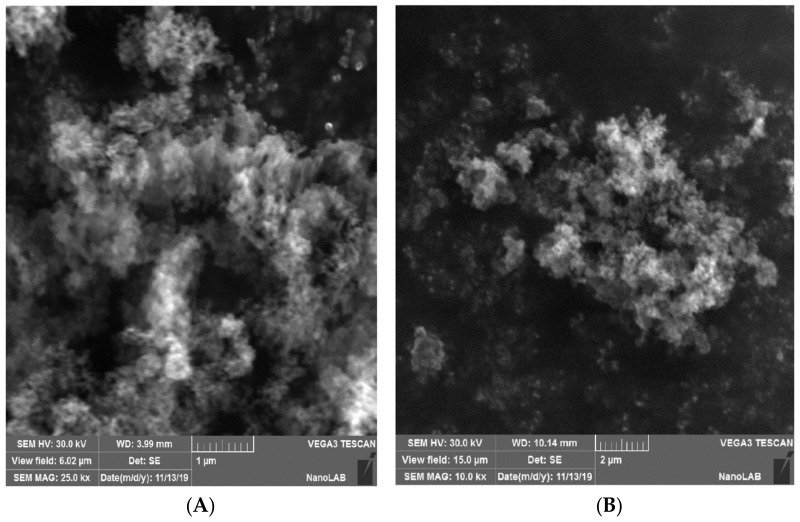
SEM images of the SiO_2_ nanoparticles (**A**) magnification 25 k× and(**B**) magnification 10 k×.

**Figure 6 materials-13-05727-f006:**
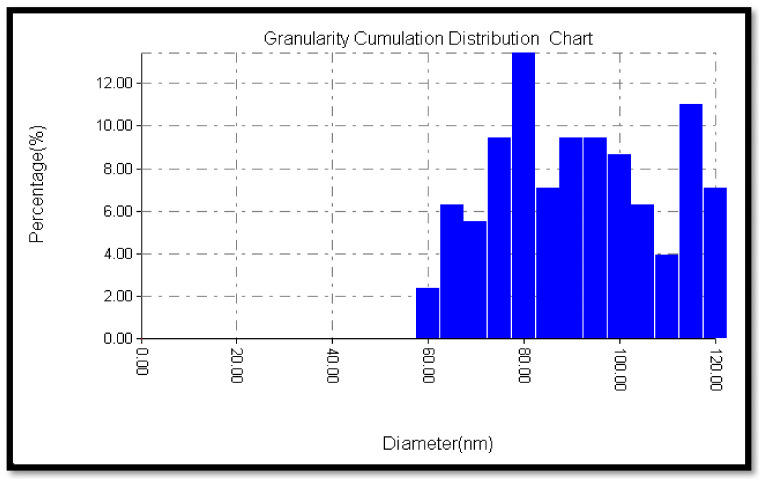
Particle size distribution of the SiO_2_ nanoparticles.

**Figure 7 materials-13-05727-f007:**
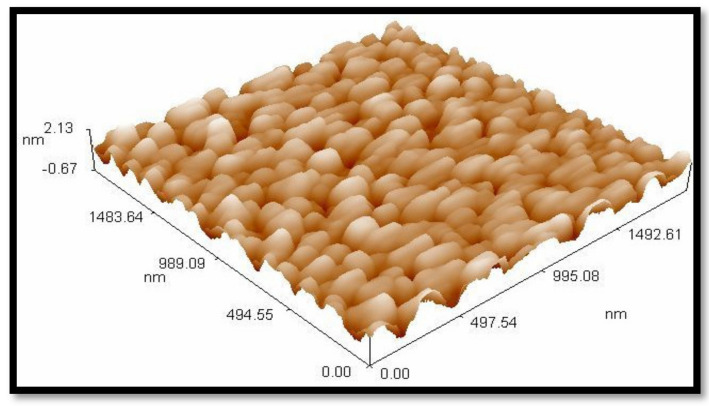
Atomic force microscopy (AFM) three-dimensional (3D) surface profile of nanosilica.

**Figure 8 materials-13-05727-f008:**
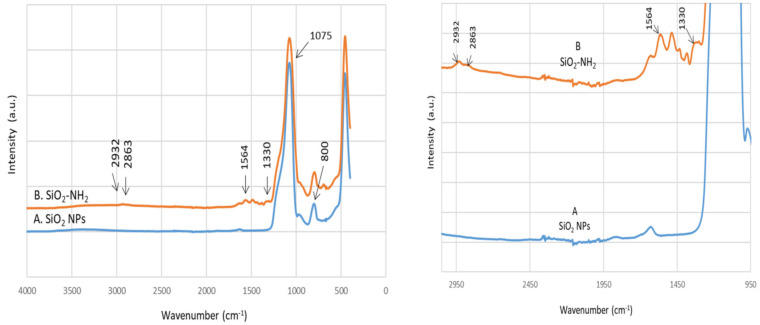
FTIR spectra of the (**A**) NS and(**B**) NS-APTES.

**Figure 9 materials-13-05727-f009:**
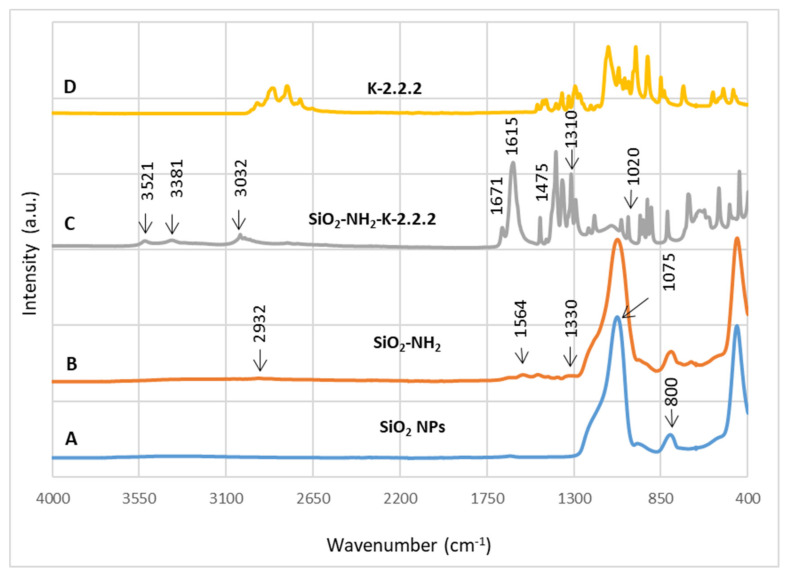
FTIR spectra of the (**A**) NS, (**B**) NS-APTES, (**C**) NS-APTES-K2.2.2, and (**D**) pure K2.2.2.

**Table 1 materials-13-05727-t001:** Determination of scandium with UV spectroscopy after extraction and stripping.

Sc, ppm Model Solution	Sc, ppm (Aqueous Phase) After Extraction	Extraction Capacity mg/g	Extraction%	Sc, ppm Concentration on the Solid Phase before Back Extraction/ Stripping	Sc, ppm Concentration in the Aqueous Phase after Back Extraction/ Stripping	Back Extraction/Stripping %
15	0.5	14.5	96.7	14.5	14	96.5
25	1.5	23.5	94.0	23.5	23	97.8
50	1.7	48.3	96.6	48.3	45	93.1
75	14	61	81.3	61	58	95.0

**Table 2 materials-13-05727-t002:** Chemical composition of the RHA and nanosilica analyzed by XRF.

Component (m %)	SiO_2_	Na_2_O_3_	MgO	CaO	Fe_2_O_3_	Al_2_O_3_	K_2_O	SO_3_
HCl (3N) (5 °C/min)	97.5	0.23	0.15	0.08	0.22	0.9	0.11	1.1
Nanosilica	98.7	0.004	0.05	0.02	0.05	0.2	0.003	0.7

**Table 3 materials-13-05727-t003:** Previous studies to recover scandium by solid-phase extraction (SPE).

Solid Phase	Year	References
Ampholyte resins, AFI-21 and AFI-22	1997	[51]
Ionic imprinted polymer materials IIP-PEI/SiO_2_	2014	[52]
Chitosan–silica hybrid materials DTPA–chitosan–silica and EGTA–chitosan–silica	2016	[53]
Silica sol-gel doped with ionic liquid	2016	[54]
Biochar	2017	[55]
Mesoporous silica-PAN	2017	[56]
Activated carbon and silica composites	2018	[57]
D201 resin	2020	[58]

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
