# Peer review of "Novel Hybrid Nanoparticles: Synthesis, Functionalization, Characterization, and Their Application in the Uptake of Scandium (III)Ions from Aqueous Media"

_materials, 2020, doi:10.3390/ma13245727_

Round 1
Reviewer 1 Report
This manuscript deals with the study of Synthesis, Functionalization, Characterization and further practical application of Novel Hybrid Nanoparticles based on natural amorphous silica from Iraqi rice husk. The authors of this manuscript obtained interesting results and conclusions.
The authors described the target of the paper, along with the practical applications of it in a clear manner.
Questions:
- Please provide a schematic representation on how the process of HNP synthesis
- Authors explained an Sc (III) recovering process basing on Table 2 data. Please point out which chemical composition information provide such results.
- Line 284: Their deformation vibrations…..Please clarify "their".
- Do authors determined an effect of APTES content on surface modification of SiO2 nanoparticles in the manuscript?
- Could the authors please explain the colloidal stability of synthesized HNP for further using?
Based on the above points, I would propose accept the manuscript after major revision.
Author Response
Thank you for your efforts to improve our manuscript! The answers are presented in the attached document and also in the revised manuscript!

Reviewer 2 Report
This study developed a new supramolecular hydride nanoparticle that can separate and recover scandium metal ions, in an aqueous phase based on molecular recognition technology. finding the efficacy of its validation in leachates from 81.3 to 96.7%, and also more complex metal values ​​and can be recovered in a concentration of 0.1 mol / L HCL.
It is undoubtedly a novel investigation, however, I consider that the document should be attended to and improved if before continuing with its processing, although it has a clear and solid methodology, it does not manifest a further discussion, the results should be discussed and compared with another similar or that it eliminates the same metal in the system, in leachate for example used as a test medium, and to highlight the advantages of this new one, a complementary table where similar studies will be shown and would allow a better discussion could facilitate the interpretation of the problems since I consider that they are important for the scientific world.
Author Response

(The authors gave the same response as above.)

Reviewer 3 Report
The article describes hybrid nanoparticles obtained from rice husk and APTES and K2.2.2 and their application in scandium adsorption and its recovery. The material was used for the solid-liquid extraction of the cations; deep physiochemical characterization of the nanoparticles was provided in the article.
The introduction provided sufficient information for the research background. Synthesis methodology is well presented and illustrated in graphs. Adsorption and recovery studies well described. Material characterization is coherent and conclusive.
One aspect that is missing is a comparison of the investigated parameters with other materials. I would recommend to add a small section regarding that topic.
Besides that, I have no reservations and recommend the article for publication.
Author Response

(The authors gave the same response as above.)

Round 2
Reviewer 1 Report
The authors answered my questions in a clear manner.